# Adaptation and Psychometric Properties of an Attitude toward Artificial Intelligence Scale (AIAS-4) among Peruvian Nurses

**DOI:** 10.3390/bs14060437

**Published:** 2024-05-23

**Authors:** Wilter C. Morales-García, Liset Z. Sairitupa-Sanchez, Sandra B. Morales-García, Mardel Morales-García

**Affiliations:** 1Escuela de Posgrado, Universidad Peruana Unión, Lima 15457, Peru; wiltermorales@upeu.edu.pe; 2Facultad de Teología, Universidad Peruana Unión, Lima 15457, Peru; 3Sociedad Científica de Investigadores Adventistas, SOCIA, Universidad Peruana Unión, Lima 15457, Peru; 4Club de Conquistadores, Orión, Universidad Peruana Unión, Lima 15457, Peru; 5Escuela Profesional de Psicología, Facultad de Ciencias de la Salud, Universidad Peruana Unión, Lima 15457, Peru; lisetsairitupa@upeu.edu.pe; 6Escuela Profesional de Medicina Humana, Facultad de Ciencias de la Salud, Universidad Peruana Unión, Lima 15457, Peru; 100097321@cientifica.edu.pe; 7Unidad de Salud, Escuela de posgrado, Universidad Peruana Unión, Km 19, Carretera Central, Lima 15033, Peru

**Keywords:** artificial intelligence, attitudes, psychometric evaluation, healthcare, technology, attitude of health personnel

## Abstract

Background: The integration of Artificial Intelligence (AI) into various aspects of daily life has sparked growing interest in understanding public attitudes toward this technology. Despite advancements in tools to assess these perceptions, there remains a need for culturally adapted instruments, particularly in specific contexts like that of Peruvian nurses. Objective: To evaluate the psychometric properties of the AIAS-4 in a sample of Peruvian nurses. Methods: An instrumental design was employed, recruiting 200 Peruvian nurses. The Attitude toward Artificial Intelligence in Spanish (AIAS-S), a cultural and linguistic adaptation of the AIAS-4, involved data analysis using descriptive statistics, confirmatory factor analysis (CFA), and invariance tests. Results: The Confirmatory Factor Analysis (CFA) confirmed a unidimensional factor structure with an excellent model fit (χ^2^ = 0.410, df = 1, *p* = 0.522, CFI = 1.00, TLI = 1.00, RMSEA = 0.00, SRMR = 0.00). The scale demonstrated high internal consistency (α = 0.94, ω = 0.91). Tests of invariance from configural to strict confirmed that the scale is stable across different demographic subgroups. Conclusions: The AIAS-S proved to be a psychometrically solid tool for assessing attitudes toward AI in the context of Peruvian nurses, providing evidence of validity, reliability, and gender invariance. This study highlights the importance of having culturally adapted instruments to explore attitudes toward emerging technologies in specific groups.

## 1. Introduction

Artificial Intelligence (AI) has marked a milestone in the modern era, standing out for its ability to simulate human cognitive processes in machines and software, with applications ranging from health to education, promising to revolutionize our existence [1,2,3]. As AI integrates into our daily lives, products like Siri and Alexa, along with the development of social robots, showcase its potential to improve quality of life, offering everything from safer driving to more efficient medical care [4,5]. However, this expansion brings significant challenges, such as the risk of job displacement and ethical concerns, reflecting a spectrum of public opinions ranging from acceptance to anxiety [6,7,8,9].

Despite the benefits AI can bring, it is not without faults and requires expert human supervision to mitigate unpredictable errors and biases. It is crucial that users, especially students, are informed about the opportunities and ethical dilemmas associated with responsible and critical use [10,11]. The need for appropriate regulations becomes evident for its integration into society and governmental processes, facing challenges in terms of scope and impact on innovation [12]. The widespread adoption of AI has also been influenced by the COVID-19 pandemic, increasing emotional and psychological dependency on these technologies. In areas like mental health and financial advising, service chatbots have shown potential benefits, though they also present risks such as social withdrawal and addiction [13,14,15,16].

In the debate over AI, concerns about the development of autonomous weaponry and the existential impact of advanced AI have been fueled by prominent figures, reflecting the need for ongoing reflection on humanity’s future with AI [17,18,19,20]. Research on public perceptions is mixed, highlighting both concerns and acknowledgment of its potential for innovative solutions [21,22]. Personality and trust emerge as crucial factors in shaping attitudes toward AI. Studies indicate that traits such as openness and conscientiousness can influence technological acceptance, while trust in AI and in the corporations developing it plays a significant role in the perception of its risk and utility [23,24,25,26].

The integration of AI into clinical practice represents a significant advancement in healthcare. This progress is reflected in the attitude and perception toward AI by healthcare professionals and nursing students, who play a crucial role in the adoption and application of these technologies. Despite the growing importance of AI, previous studies have identified a notable lack of knowledge and understanding about AI in the nursing context, especially in regions like Jordan, recognized as a center of medical care and medical tourism with steady annual growth in foreign patients [27,28].

The attitude toward AI in the clinical community varies, with studies indicating that, although there is a generally positive attitude toward its use, there is a significant gap in knowledge and practical experience with these technologies [29,30]. This challenge extends to nursing education, where more than 70% of nurses and nursing students acknowledge AI’s potential to revolutionize healthcare, but admit to a limited understanding of its practical application [31]. Anxiety and lack of confidence in using AI are also highlighted issues, affecting future professional decisions and openness to specialization in fields like radiology [32].

Furthermore, interdisciplinary collaboration and the gathering of information from a diverse group of healthcare students are essential for the effective incorporation of AI into modern medicine, highlighting the need for a comprehensive approach that includes training in data acquisition and protection, AI ethics, and critical evaluation and interpretation of AI applications in health [33,34]. As the field of AI in healthcare continues to evolve, it is imperative to develop and enhance nurses’ competencies to adapt to these technological changes, ensuring they are equipped with the necessary knowledge to lead and shape the future of nursing practice in the AI era [35,36].

Gender differences in attitudes toward AI have been observed across various domains, including education, healthcare, and professional environments. For instance, a multiple-group analysis of prospective German teachers revealed gender-specific factors that influence the acceptance of AI-based applications, underscoring the importance of addressing these aspects in educational settings [37]. Similarly, attitudes toward AI among medical and pharmacy students have shown variations that could affect future implementation and use in medical practice [38,39]. In the healthcare sector, a study among Chinese dermatologists indicated varying levels of engagement with AI, influenced by gender among other factors, which could impact the adoption and effective use of AI technologies in dermatology [40]. Furthermore, exploring attitudes toward AI in broader demographic samples, including different age groups and cultural backgrounds, further complicates the landscape, suggesting that these attitudes are not only dependent on gender but also on a multitude of sociodemographic factors [41].

This complex landscape underscores the importance of developing accurate and reliable assessment tools to capture general attitudes toward AI, addressing the need for multidimensional approaches in its study [42,43,44,45,46,47]. In this regard, the evolution in the development of instruments to assess attitudes toward artificial intelligence (AI) has been significant over the last decade, with notable contributions such as the General Attitudes toward Artificial Intelligence Scale (GAISS) [42], the Attitude toward Artificial Intelligence (ATAI scale) [43], and the Threats of Artificial Intelligence Scale (TAI) [48]. These studies have established a foundation for measuring perceptions and attitudes toward AI, highlighting both the positive aspects and the fears associated with its implementation and development. However, despite these advancements, significant limitations in these scales have been identified, justifying the need for further studies, especially in specific cultural and geographical contexts like Peru. The GAISS, for example, although comprehensive, may be impractical for large-scale applications due to its length, while the ATAI, being more concise, might not capture the complexity and nuances of attitudes toward AI, focusing on extremes that may not reflect the intermediate perceptions of individuals. Moreover, the TAI specifically focuses on fears related to AI, which might not provide a complete picture of the general attitudes toward this technology.

The study by Grassini [49] on the Attitude toward AI Scale (AIAS-4) represents a significant advance by offering a brief and psychometrically validated measure that addresses some of these limitations. However, it is crucial to recognize that attitudes toward AI are neither static nor universal. The rapid evolution of AI technology, along with the emergence of large language models like Chat Generative Pre-trained Transformer (ChatGPT), has transformed the way people interact with and understand AI [46]. This technological dynamism, combined with the cultural, social, and economic particularities of each region, suggests that instruments developed and validated in one context may not be directly applicable or fully relevant in another, such as in Peru [50,51]. Therefore, it is essential to have scores derived from measurement instruments that prove to be valid and reliable within the specific cultural context of Peru. Thus, the aim of this study is to evaluate the psychometric properties of the scores obtained through a brief scale that measures the general attitude toward artificial intelligence among Peruvian nurses.

## 2. Methods

### 2.1. Design and Participants

This study, of an instrumental nature [52], was based on convenience sampling for participant selection. The specific inclusion criteria regarding attitude toward Artificial Intelligence were as follows: currently employed as nurses, having at least one year of experience in the nursing field to ensure basic familiarity with health technologies, and having had some previous interaction with AI-based tools or systems in their work environment. To determine the necessary sample size, an electronic sample size calculator was used, following the recommendations of Soper [53]. This calculation took into account several critical factors, including the number of observed and latent variables in the proposed model, an expected moderate effect size (λ = 0.10), a statistical significance level of α = 0.05, and a statistical power of 1 − β = 0.80. Although the minimum required sample size was estimated at 85 participants, a total of 243 nurses were recruited, with ages ranging from 22 to 62 years (M = 34.93, SD = 7.94). The majority of participants were women (63.0%), single (53.5%), held a bachelor’s degree (53.1%), and were employed under Contractual Services Administration (CAS) contracts, a form of temporary employment used in the public sector (26.7%) (Table 1).

### 2.2. Instruments

General Attitude toward Artificial Intelligence. The Attitude toward Artificial Intelligence Scale (AIAS) assesses public attitudes toward artificial intelligence (AI) [49]. This scale, validated in the UK and USA, consists of four items that capture individual beliefs about the influence of AI on their lives, careers, and humanity in general [49]. The AIAS is structured around a single dimension, reflecting a composite attitude toward AI, including perceived utility and potential impact on society. The scale demonstrated adequate internal consistency, with a Cronbach’s alpha of 0.902 and a McDonald’s omega of 0.904. It employs a 10-point Likert scale (1 = Not at all, 10 = Completely agree).

The Spanish adaptation of the AIAS was carried out using a rigorous cultural adaptation method [54] to ensure linguistic accuracy and conceptual equivalence with the original instrument. It is important to note that permission was obtained from the original author for the use of the instrument. The adaptation procedure included the following stages:Two bilingual translators, native Spanish speakers, independently performed the initial translation of the AIAS into Spanish. Subsequently, both versions were compared to create an initial unified version.This translated version was then back-translated into English by two native English speakers from the United States, competent in Spanish but without prior knowledge of the AIAS. This stage aimed to confirm the preservation of the original meaning in the translation.An expert panel, consisting of two psychologists and two nurses, examined the Spanish version along with the back-translated English versions, with the purpose of developing a preliminary version of the AIAS in Spanish (AIAS-S).This preliminary version was subjected to the evaluation of a focus group composed of 10 nurses, to verify its comprehension and readability. The issues identified at this stage led to the making of relevant linguistic adjustments, resulting in the final version of the instrument in Spanish, called AIAS-S, which translates to “Attitude toward Artificial Intelligence in Spanish” (see Table 2).

### 2.3. Procedure

The study was conducted under strict ethical principles, having received approval from the Ethics Committee of the Peruvian university, under the reference code 2023-CEUPeU-033. Permission was requested and obtained from the administration of the involved hospitals, thus ensuring institutional collaboration and adequate access to participants in the hospital setting. The privacy and confidentiality of the participants’ information were guaranteed, ensuring the acquisition of their informed consent before proceeding with the survey application. The questionnaire administration took place in person at two major Peruvian universities, emphasizing the voluntary and anonymous nature of participation.

### 2.4. Data Analysis

A preliminary descriptive analysis of the AIAS-S items was conducted, including the evaluation of the mean, standard deviation, skewness, and kurtosis, along with a corrected inter-item correlation analysis. Skewness (g1) and kurtosis (g2) values within the range of ±1.5 were considered acceptable [55]. Additionally, corrected item-total correlation analysis was employed to discard any items with a r(i − tc) ≤ 0.2 or in cases of multicollinearity [56].

We proceeded with a Confirmatory Factor Analysis (CFA) for the scale, using the Maximum Likelihood Robust (MLR) estimator, which is suitable for data presenting deviations from normality [57]. The indices used to evaluate the model fit included chi-square (χ^2^), Comparative Fit Index (CFI) and Tucker–Lewis Index (TLI) (≥0.95), as well as Root Mean Square Error of Approximation (RMSEA) and Standardized Root Mean Square Residual (SRMR) (≤0.08) [56,58]. Internal consistency was verified using Cronbach’s alpha and McDonald’s omega, expecting values above 0.70 to consider it adequate [59].

A hierarchical sequence of measurement invariance models was implemented. Initially, configural invariance was analyzed, serving as the reference model. This analysis was followed by the assessment of metric invariance, which ensures the equality of factor loadings across groups, and then scalar invariance, which additionally equalizes item intercepts. To verify invariance across these models, a modeling strategy was adopted that involves observing differences in the Comparative Fit Index (CFI). According to Chen [60], ΔCFI differences less than 0.010 indicate that invariance is maintained between groups, thus validating the consistency of the scale across different gender contexts.

Statistical analyses were performed using RStudio [61] with R version 4.1.1 (R Foundation for Statistical Computing, Vienna, Austria). For the confirmatory factor analysis and structural equation modeling, the “lavaan” package was used [62], and to facilitate the analysis of measurement invariance, the “semTools” package was employed [63].

## 3. Results

### 3.1. Descriptive Statistics of Items

The results of the Attitude Scale toward Artificial Intelligence (AIAS) show variability in perceptions about AI, with average scores ranging from 5.84 to 6.73 on a 10-point scale. Item 3 displays the highest average (M = 6.73, SD = 2.65), indicating a positive expectation toward future use of AI. Conversely, Item 1 has the lowest average (M = 5.84, SD = 2.62), reflecting a potentially more cautious view on the immediate personal benefits of AI. All items exhibit skewness (g1) and kurtosis (g2), suggesting a slight deviation from normality, but not excessively so. The item-total correlations (r.cor) for each item are high, all around 0.85 or more, demonstrating a strong relationship between each item and the total score of the scale. This implies that each item significantly contributes to the overall measurement of attitudes toward AI, and therefore, it is not recommended to eliminate any. Moreover, the correlation matrix between items reveals high coefficients, ranging from 0.75 to 0.86, confirming that perceptions of AI are consistently assessed across the different statements of the scale (Table 2).

### 3.2. Confirmatory Factor Analysis

A Confirmatory Factor Analysis (CFA) was conducted following the guidelines established by Grassini [49]. The evaluation of the resulting model indicated a significant improvement in the fit indices: χ^2^ = 0.410, gl = 1, *p* = 0.522, CFI = 1.00, TLI = 1.00, RMSEA = 0.00 (90% CI 0.00–0.13), SRMR = 0.00. The factor loadings were of adequate magnitudes, all surpassing the threshold of 0.50, reinforcing the validity of the construct measured by the scale (Figure 1). These results corroborate the unidimensional factor structure of the scale and its appropriateness for assessing the general attitude toward Artificial Intelligence. 

### 3.3. Reliability

In the analysis of internal consistency of our scale, the results were highly satisfactory. The reliability coefficient for Cronbach’s alpha (α) was 0.94 and for McDonald’s omega (ω) was 0.91.

### 3.4. Measurement Invariance

Starting with configural invariance, which establishes a baseline model without constraints, a high CFI of 0.999 was observed, indicating excellent fit of the initial model. Upon introducing metric invariance, which involves equality in factor loadings across groups, the CFI remained perfect at 1.000, with a ΔCFI of −0.001, demonstrating that the scale measure constructs equivalently across genders. Progressing to scalar invariance, where both factor loadings and intercepts are equalized, the CFI slightly decreased to 0.995 with a ΔCFI of 0.005. Although this change exceeds the threshold of 0.001, it remains below 0.01, suggesting still good measurement equivalence between groups. Finally, strict invariance, which adds equality of error variances to the previous constraints, resulted in a CFI of 0.998 with a ΔCFI of −0.003, showing improvement from the scalar model (Table 3).

## 4. Discussion

Artificial Intelligence (AI) has revolutionized various sectors, including healthcare and education, by simulating human cognitive processes in machines and software. This integration has led to improvements in quality of life but also presents significant challenges such as job displacement and ethical dilemmas. As AI becomes more prevalent in our daily lives, human oversight becomes crucial to mitigate errors and biases. Additionally, its rapid adoption has been driven by circumstances such as the COVID-19 pandemic, increasing our emotional and psychological dependence on these technologies. In the healthcare context, particularly in nursing, although there is generally a positive perception of AI, a notable gap in knowledge and practical experience is observed. This underscores the importance of training in the use of AI, critically evaluating its applications in health. Despite advances in developing instruments to assess attitudes toward AI, there are limitations that justify further research, especially in specific contexts like Peru, to develop culturally relevant evaluation tools. This study aims to evaluate the psychometric properties of an attitude scale toward AI, specifically adapted for Peruvian nurses, addressing the need for precise and reliable evaluation tools in this rapidly evolving field.

The Confirmatory Factor Analysis (CFA) for the AIAS-S was conducted following the guidelines of Grassini [49]. Comparing our findings with existing literature, we observed both similarities and significant variations. For instance, Grassini [49] reports a good model fit for the AIAS-4, similar to our study, which supports the proposal of a unifactorial structure for measuring attitudes toward AI in various contexts. However, compared to other studies such as those by Schepman and Rodway [42] and Kieslich et al. [48], who explored more complex and varied factorial structures in their respective studies, significant differences are observed. Schepman and Rodway [42] validated a two-factor model for the GAAIS, while Kieslich et al. [48] reported a significant model with good fit indices for the TAI. These discrepancies highlight the diversity in perceptions and attitudes toward AI and underscore the need to adapt and assess measurement tools specific to each cultural and linguistic context. Although our study identified a unifactorial structure, we advise caution in generalizing this homogeneity to other contexts. Given that the Spanish-speaking environment may be influenced by factors such as exposure to technology, the level of understanding of AI, and specific sociocultural variables that modulate the perception of its benefits and risks. This observation invites further reflection on how cultural and contextual differences can affect factorial structures in assessing attitudes toward artificial intelligence.

Similarly, the AIAS-S follows a line similar to the research conducted by Grassini [49], Schepman and Rodway [42], and Kieslich et al. [48], each of which addressed different dimensions and contexts of attitude toward AI. Comparing the reliability findings of our study with those mentioned above, there is a general consistency in the good internal reliability of the scales used, with Cronbach’s alpha coefficients generally indicating acceptable to good internal consistency. In this regard, Grassini’s [49] research established good internal consistency for the AIAS-4, a finding that aligns with our study and supports the reliability of the instrument for measuring attitudes toward AI. Similarly, Schepman and Rodway [42] and Kieslich et al. [48] reported Cronbach’s alpha values reflecting satisfactory reliability in their respective scales, highlighting the robustness of the instruments to capture variations in attitudes and perceptions toward AI. These findings are crucial as they ensure that measurements of attitudes toward AI are stable and comparable across different studies and populations. 

Measurement invariance in the AIAS-4 was analyzed across four levels (configural, metric, scalar, and strict) with results indicating exceptional fit in all models. Comparative Fit Index (CFI) values ranged from 0.995 to 1.000, with minimal variations in ΔCFI, suggesting that the AIAS scale consistently measures the same construct among both men and women, which is essential for making valid comparisons between these groups. Finally, the consistency in reliability observed across different studies suggests that attitudes toward AI can be effectively conceptualized and measured using well-designed scales. This indicates a coherent underlying structure in how people evaluate AI, enabling meaningful comparisons of attitudes across various cultural and demographic contexts.

### 4.1. Implications

The integration of Artificial Intelligence (AI) into professional practice and healthcare systems represents a promising technological advancement with the potential to revolutionize how healthcare is delivered and managed. Assessing the psychometric properties of the AIAS-S among Peruvian nurses provides an empirical foundation for understanding healthcare professionals’ receptiveness and perceptions toward this emerging technology. The findings suggest significant implications for professional practice, health policy, and theoretical development in the field of health AI.

From a practical perspective, the generally positive attitude toward AI identified among Peruvian nurses indicates an openness to incorporating these technologies into their daily practice. This suggests the need to develop training and professional development programs that focus not only on the technical skills required to operate AI technologies but also on understanding their ethical, regulatory, and practical applications in the healthcare setting.

Regarding health policies, the results underline the importance of formulating regulatory frameworks that support the safe and ethical implementation of AI in healthcare. Policies should focus on ensuring patient data privacy and security while promoting equity in access to advanced healthcare technologies. Moreover, it is crucial that these policies encourage interdisciplinary collaboration among engineers, AI designers, healthcare professionals, and patients to ensure that AI solutions are relevant, clinically sensitive, and culturally appropriate.

From a theoretical perspective, this study contributes to the existing body of knowledge by providing empirical evidence about nurses’ attitudes toward AI in a specific context. This suggests the need to continue exploring how cultural, contextual, and educational factors influence healthcare professionals’ perceptions of AI. Future research could aim to examine attitudes toward AI across different medical specialties and geographical contexts to better understand the variables that facilitate or inhibit the adoption of AI in healthcare practice.

### 4.2. Limitations

One of the main limitations of our study is the use of a cross-sectional design. The cross-sectional nature of the study prevents the assessment of how attitudes toward AI may change over time, especially in a field as dynamic and rapidly evolving as AI technology. Another limitation is the potential for social desirability bias in the participants’ responses. Given the context of the study, the nurses might have responded in a way that reflects more positive perceptions of AI, influenced by the perception of AI as a valuable technological innovation in the healthcare field. Although measures were taken to ensure confidentiality and the importance of honest responses was emphasized, the inherent self-assessment in surveys may still be subject to this type of bias. Additionally, the absence of studies on convergent and discriminant validity is identified as a limitation. Convergent validity studies with other relevant scales for assessing attitudes toward AI, such as the General Attitudes toward Artificial Intelligence Scale (GAISS) and the Threats of Artificial Intelligence Scale (TAI), would have strengthened the interpretation of the AIAS-4 results. On the other hand, discriminant validity studies with scales measuring conceptually different variables, such as the Technology Anxiety Scale, could have helped establish the specificity of the AIAS-4 in measuring attitudes toward AI, contrasting with other psychological constructs. It is advisable that future research includes these types of validation to confirm the robustness of the AIAS-4 and its ability to accurately measure attitudes toward AI without being influenced by related but distinct constructs.

## 5. Conclusions

The present study marks a significant advancement in understanding attitudes toward Artificial Intelligence (AI) within the specific context of Peruvian nurses, valuably contributing to the body of knowledge at the intersection of technology and health. By evaluating the psychometric properties of the scores obtained with the AIAS-S, which was culturally and linguistically adapted for the Peruvian context, this study demonstrated that the scores are valid and reliable. The findings, revealing a coherent factorial structure and robust internal reliability in the AIAS-S scores, align with previous research in other contexts, highlighting the importance of cultural considerations in psychometric assessment.

## Figures and Tables

**Figure 1 behavsci-14-00437-f001:**
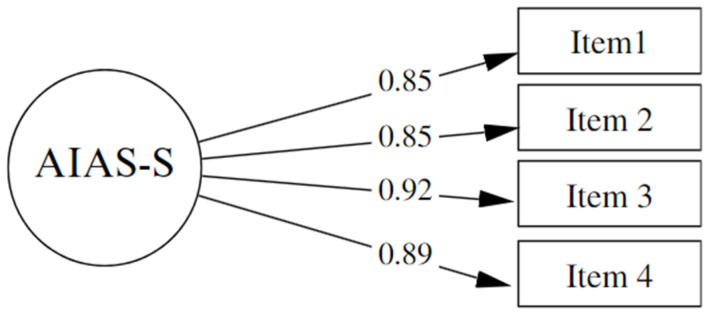
Path Diagram of the Confirmatory Factor Analysis Model for the AIAS-4 Scale.

**Table 1 behavsci-14-00437-t001:** Descriptive Statistics.

Characteristics		n	%
Gender	Female	153	63.0
	Male	90	37.0
Marital Status	Married	80	32.9
	Cohabiting	20	8.2
	Living together	13	5.3
	Divorced	130	53.5
	Widowed	54	22.2
Level of Education	Specialty	129	53.1
	Bachelor’s Degre	60	24.7
	Postgraduate	65	26.7
Employment Status	Contract (CAS)	36	14.8
	Permanent Contract)	68	28.0
	Tenured	16	6.6
	Substitute	58	23.9
	Third-party	153	63.0

**Table 2 behavsci-14-00437-t002:** Descriptive Statistics and the polychoric correlation matrix.

English Version	Spanish Version	M	sd	g1	g2	r.cor	1	2	3	4
1. I believe that AI will improve my life	1. Creo que la IA mejorará mi vida	5.84	2.62	−0.07	−0.85	0.86	-			
2. I believe that AI will improve my work	2. Creo que la IA mejorará mi trabajo	5.87	2.77	−0.07	−1.02	0.86	0.86 **	-		
3. I think I will use AI technology in the future	3. Pienso que usaré tecnología de IA en el futuro	6.73	2.65	−0.41	−0.82	0.85	0.78 **	0.78 **	-	
4. I think AI technology is positive for humanity	4. Pienso que la tecnología IA es positiva para la humanidad	6.16	2.61	−0.1	−0.87	0.83	0.75 **	0.76 **	0.82 **	-

** = *p* < 0.01.

**Table 3 behavsci-14-00437-t003:** Invariance according to sex.

Invariance	χ^2^	df	*p*	TLI	RMSEA	SRMR	CFI	ΔCFI
Configural	2.228	2	0.328	0.996	0.031	0.004	0.999	
Metric	2.869	5	0.72	1.013	0.000	0.013	1.000	−0.001
Scalar	9.862	8	0.275	0.993	0.044	0.023	0.995	0.005
Strict	12.764	12	0.386	0.998	0.023	0.028	0.998	−0.003

## Data Availability

Data can be provided at the request of the corresponding author.

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
