# Peer review of "Adaptation and Psychometric Properties of an Attitude toward Artificial Intelligence Scale (AIAS-4) among Peruvian Nurses"

_behavsci, 2024, doi:10.3390/bs14060437_

Round 1
Reviewer 1 Report
Comments and Suggestions for Authors
You worked hard to write this research.
I think that the flow and methodology of the paper appear to be appropriate.
However, important validation procedures in conducting psychometric measures were omitted in this study.
- 1. Reliability verification: The reliability of the instrument could vary depending on the number of items, but this aspect was overlooked in the study.
- 2. Validity verification: Both construct validity and criterion validity were missing. (Referring to various tool development books might be helpful. For example, "Measurement in Nursing and Health Research" by Waltz.)
Was permission obtained from the tool developer, Grassini? It seems that this aspect was not addressed in the study.
According to the results of validity verification, the discussion content is likely to change.
Thank you.

Author Response
I think that the flow and methodology of the paper appear to be appropriate.
However, important validation procedures in conducting psychometric measures were omitted in this study.
- Reliability verification: The reliability of the instrument could vary depending on the number of items, but this aspect was overlooked in the study.
Response: Thank you for your comments, however to clarify this part, the reliability section was added to clarify this part.
- Validity verification: Both construct validity and criterion validity were missing. (Referring to various tool development books might be helpful. For example, "Measurement in Nursing and Health Research" by Waltz.)
Response: Thank you for your comments and suggestions. The current study primarily focused on the linguistic and cultural adaptation of the Scale of Attitudes towards Artificial Intelligence (AIAS-S) and its structural validation through a Confirmatory Factor Analysis (CFA). In addition, measure invariance by gender was assessed, restricting the analysis to a sample of Peruvian nurses. This approach is justified by the need to initially establish the structural validity and reliability of the adapted version of the instrument, which are fundamental steps in the process of adapting psychometric tools (Borsa, Damásio, & Bandeira, 2012). Regarding construct and criterion validation, it is essential to clarify that construct validity was partially addressed through the CFA, which helps confirm that the scale's factorial structure is consistent with the underlying theory (Kline, 2023). However, we acknowledge the absence of a criterion validity analysis, which would involve correlating AIAS-S results with other established measures to assess similar or related concepts (DeVellis, 2016). Nevertheless, this study focused on structural validity due to the initial scope limitations of the project. Therefore, it is indicated in the limitations that these analyses were not included in the current phase and could be considered for future research. In this sense, this initial study seeks to establish a foundational base upon which more comprehensive evaluations can be conducted in the future, expanding the scope to include convergent and divergent validation. These steps are essential to confirm the utility of the AIAS-S in various situations and to compare its results with other existing tools in the field of AI psychometry.
- Was permission obtained from the tool developer, Grassini? It seems that this aspect was not addressed in the study.
Response: We obtained Grassini's permission, so it was added in the instrument section, stating that the author's permission was obtained.
According to the results of validity verification, the discussion content is likely to change.
Thank you.
- Authors worked hard to write this research. Was permission obtained from the tool developer, Grassini? It seems that this aspect was not addressed in the study.
Response: It was stated above, see previous answer
- Contents Comments Title How about changing the title? Translation and Psychometric properties of an artificial intelligence attitude scale (AIAS-4)
Response: Thank you for your comments, we changed the title specifying the population at your suggestion
- Instruments 1. Reliability verification: The reliability of the instrument could vary depending on the number of items, but this aspect was overlooked in the study. 2. Validity verification: Both construct validity and criterion validity were missing. (Referring to various tool development books might be helpful. For example, "Measurement in Nursing and Health Research" by Waltz.)
Response. The information for this suggestion is provided in question 3.
- Procedure The content regarding procedure seems more suitable for ethical considerations.
- Equation model fit?
Response: The accuracy of the figure description was made
Discussion According to the results of validity verification, the discussion content is likely to change.
The pdf is attached with the suggested changes in red

Reviewer 2 Report
Comments and Suggestions for Authors
Minor revisions required according to information in the revision report.

Author Response
specific comments
Line 2 - Title: should include the population for which it has been validated. Suggestion: Translation and Psychometric Properties of an Attitude towards Artificial Intelligence Scale (AIAS-4) for Peruvian Nurses;
Response: Thank you for your comments, we made the suggested changes.
Line 25 - Abstract: the author uses the acronym of AIAS for the first time (its meaning should be put in parentheses);
Response: Thank you for your comments, we made the suggested changes.
Line 26 – Abstract: I suggest only the use of the acronym CFA as it has already been identified in the previous line;
Response: Thank you for your comments, we made the suggested changes.
Line 34 - Keywords: instead of “attitudes” I suggest "Attitude of Health Personnel" and instead of “nurse” it can be “nurses”
Response: Thank you for your comments, we made the suggested changes.
Line 104 - Introduction: - Put "Chat Generative Pre-trained Transformer" before the acronym ChatGPT
Response: Thank you for your comments, we made the suggested changes.
Note on the introduction: The final part of the introduction contains information about the research problem, but it should also be clear about the aim of the study and main question/questions?
Response: Thank you, however we have decided not to add the question, since research methodology, especially in psychometric studies that involve the validation of scales or instruments, is more common to focus on clear and specific objectives than on formulating explicit research questions.
In this sense, the structure and focus of the study are oriented towards objectives of "evaluating the validity and reliability of the scale..." rather than research questions, because these objectives already imply the need to perform specific statistical analyzes that will confirm or they will refute the usefulness of the instrument in the desired context. Furthermore, in the psychometric literature, the validation of an instrument is seen as a necessity to ensure that the measurements are accurate and representative of the construct they are intended to measure (Clark & Watson, 1995; DeVellis, 2016).
Clark, L. A., & Watson, D. (2016). Constructing validity: Basic issues in objective scale development.
Response: Thank you for your comments, we made the suggested changes.
DeVellis, R. F., & Thorpe, C. T. (2021). Scale development: Theory and applications. Sage publications.
Response: Thank you for your comments, we made the suggested changes.
Line 124 - Methods: - The reference "[46]" before the final dot?;
Response: Thank you for your comments, we made the suggested changes.
Line 132 - before table 1 in previous text, there should be some reference for consulting it.
Response: Thank you for your comments, we made the suggested changes.
Line 137 - when mention that the scale was validated in the USA and UK, what is the source of the information?
Response: Thank you for your comments, we made the suggested changes.
Line 149 to 170 - seems to repeat information. Consider better organising and clarifying the different phases?
Response: Thank you for your comments, we made the suggested changes.
Lines 188-193 - put acronyms according to the rules? e.g. Comparative Fit Index (CFI) and Tucker-Lewis Index (TLI)
Response: Thank you for your comments, what was indicated was added
Response: Thank you for your comments, the acronyms in the methodology were clarified therefore they are not added in this section
Note on the methods: clear, complying with ethical issues and specifying the methodology used to analyse, translate and validate the scale, as well as mentioning the software used.
Lines 222 and 223 – Results: see acronyms? RMSEA e.g. (Root Mean Square Error of Approximation); SRMR (Standardized Root Mean Square Residual).
Response: Thank you for your comments, the acronyms in the methodology were clarified therefore they are not added in this section
Note on the results: The results are interpreted appropriately, in a well-founded way and appear to be significant. The figures and tables are appropriate, easy to understand and show the data correctly.
Lines 233 to 248 - The beginning of the discussion repeats ideas already set out in the introduction. It could be better organised by introducing a more specific relationship with the population studied and the results obtained...
Response: Thank you for your comments, we made the suggested changes.
Note on the discussion: It focuses on important aspects, comparing them with important results from other studies. The overall discussion seems to be balanced and coherent.
Information about the implications of IA is important, particularly with regard to training health professionals in how to use it more correctly.
Response: Thank you
References:
In reference 46 try to put the year or link for consultation?
Response: Thank you for your comments, we made the suggested changes.
The document is attached in PDF, highlighting the suggestions indicated in red throughout the document.

Reviewer 3 Report
Comments and Suggestions for Authors
Dear editor
The article has a relatively simple objective, limited to translating the items of a scale - that is too short (4 items), and carrying out a confirmatory factorial analysis. There would be a lot to gain from adding some additional statistical analyses, for example, evaluating any differences in the mean (M) and standard deviation values (SD) of the responses to the items according to the criterion variables gender, marital status, level of education, employment status, as well as some convergent validity studies (with other scales for assessing attitudes towards AI cited in the literature review) and divergent validity studies in relation to scales relating to conceptually different variables.
Further validity studies would also be justified in order to present the psychometric attributes of the scale in comparison with other existing scales.
In this sense, the article could be improved to justify subsequent publication.
Specifically with regard to the text, a few changes are suggested:
1) On line 129, the APA standard for presenting M and SD values should be adopted, with two decimal places for the mean (M) and one for the standard deviation (SD), harmonizing with the presentation of M and SD values on line 202. Please correct;
2) In line 170, the word for "table" should be capitalized, as in line 216. Please correct;
3) In Table 2, in relation to the alpha value column, it is not clear whether the values refer to the alpha of the scale (if so, avoid repeating them) or to the value of the alpha with the removal of the respective item (in this case, this should be specified). Please correct;
4) There are omissions in the final bibliographical references in the following cases: reference 12, 40, 46, 50, 53. Please correct;
5) It would be desirable to standardize the presentation of the final references: a) in some cases the pages of the articles are indicated, in others they are not, b) in some cases the locality and the publisher are indicated, in others only the publisher and in others neither one nor the other, c) in some cases the title of the journal is in full and in others it is abbreviated. Please correct;
6) In line 413, a space should be inserted between "A" and "National". Please correct.
Author Response
The article has a relatively simple objective, limited to translating the items of a scale - that is too short (4 items), and carrying out a confirmatory factorial analysis. There would be a lot to gain from adding some additional statistical analyses, for example, evaluating any differences in the mean (M) and standard deviation values (SD) of the responses to the items according to the criterion variables gender, marital status, level of education, employment status, as well as some convergent validity studies (with other scales for assessing attitudes towards AI cited in the literature review) and divergent validity studies in relation to scales relating to conceptually different variables.
Further validity studies would also be justified in order to present the psychometric attributes of the scale in comparison with other existing scales.
Thank you for your detailed comments and suggestions. Indeed, the study focused on translating and validating the Spanish Attitudes towards Artificial Intelligence Scale (AIAS-S) through a Confirmatory Factor Analysis (CFA) and included the evaluation of measurement invariance by gender, using a sample limited to Peruvian nurses. The choice of this approach is justified by the need to first establish the structural validity and reliability of the new linguistic version of the scale, which is a crucial step in the process of adapting psychometric instruments (Borsa, Damásio, & Bandeira, 2012). Regarding the additional analyses you mentioned, such as differences in means and standard deviations according to demographic variables and the conduct of convergent and divergent validity studies, these are undoubtedly valuable for a deeper understanding of the dynamics and applications of the scale. However, our study was designed as an initial investigation, aiming to establish the reliability and factorial structure of the AIAS-S in a specific context (Kline, 2015). In psychometric research, it is common to start with studies that prioritize factorial structure and reliability before moving on to more complex analyses (DeVellis, 2016). That said, we recognize that including additional analyses could have enriched our findings and provided a broader view of how different demographic factors might influence attitudes toward AI. This study, therefore, lays the groundwork for future research that could include comparisons with other scales and expand the evaluation of the scale to multiple dimensions and contexts, as you suggest. In the study limitations, it is mentioned that future research could address these uncovered aspects, evaluating the scale in different populations and with a broader approach that includes convergent and divergent validation. These steps are essential to confirm the utility of the AIAS-S in various situations and to compare its results with other existing tools in the field of AI psychometry.
References:
Borsa, J. C., Damásio, B. F., & Bandeira, D. R. (2012). Adaptação e validação de instrumentos psicológicos entre culturas: algumas considerações. Paidéia (Ribeirão Preto), 22, 423-432.
Kline, R. B. (2023). Principles and practice of structural equation modeling. Guilford publications.
DeVellis, R. F., & Thorpe, C. T. (2021). Scale development: Theory and applications. Sage publications.
In this sense, the article could be improved to justify subsequent publication.
Specifically with regard to the text, a few changes are suggested:
1) On line 129, the APA standard for presenting M and SD values should be adopted, with two decimal places for the mean (M) and one for the standard deviation (SD), harmonizing with the presentation of M and SD values on line 202. Please correct;
Response: Thank you for the recommendation; the mentioned corrections have been made.
2) In line 170, the word for "table" should be capitalized, as in line 216. Please correct;
Response: Thank you for the recommendation; the mentioned corrections have been made.
3) In Table 2, in relation to the alpha value column, it is not clear whether the values refer to the alpha of the scale (if so, avoid repeating them) or to the value of the alpha with the removal of the respective item (in this case, this should be specified). Please correct;
Response: The alpha was removed since the “reliability” section was added, and the polychoric item correlations were included.
4) There are omissions in the final bibliographical references in the following cases: reference 12, 40, 46, 50, 53. Please correct;
5) It would be desirable to standardize the presentation of the final references: a) in some cases the pages of the articles are indicated, in others they are not, b) in some cases the locality and the publisher are indicated, in others only the publisher and in others neither one nor the other, c) in some cases the title of the journal is in full and in others it is abbreviated. Please correct;
6) In line 413, a space should be inserted between "A" and "National". Please correct.
Response for 4, 5, and 6: Thank you for the recommendation; corrections have been made throughout the references.
Document is attached with the changes in red for a better evaluation

Round 2
Reviewer 3 Report
Comments and Suggestions for Authors
Dear Editor
I maintain my opinion on the modest contribution of the study to the production of knowledge in the area, for two reasons: a) the scale used in the study is short (only 4 items), b) validity studies are limited to the assessment of the factorial structure of the 4 items on the scale.
In addition to this aspect, I recognize the relevance of the topic (attitudes towards AI) and I also recognize the effort to formally improve the article.
Having considered this last effort, I am in favor of publishing
Author Response
Thank you for your recommendations, which were suitable for the improvements of the document